# Early Triple Negative Breast Cancer: Conventional Treatment and Emerging Therapeutic Landscapes

**DOI:** 10.3390/cancers12040819

**Published:** 2020-03-29

**Authors:** Anna Diana, Francesca Carlino, Elisena Franzese, Olga Oikonomidou, Carmen Criscitiello, Ferdinando De Vita, Fortunato Ciardiello, Michele Orditura

**Affiliations:** 1Division of Medical Oncology, Department of Precision Medicine, School of Medicine, “Luigi Vanvitelli” University of Campania, 80131 Naples, Italy; francesca91carlino@gmail.com (F.C.); elifranzese@hotmail.it (E.F.); ferdinando.devita@unicampania.it (F.D.V.); fortunato.ciardiello@unicampania.it (F.C.); michele.orditura@unicampania.it (M.O.); 2Cancer Research UK, Edinburgh Centre, MRC Institute Genetics and Molecular Medicine, University of Edinburgh, Edinburgh EH4 2XR, UK; Olga.Oikonomidou@ed.ac.uk; 3IEO, European Institute of Oncology IRCCS, 20141 Milan, Italy; carmen.criscitiello@ieo.it

**Keywords:** triple negative breast cancer, neo-(adjuvant), biomarkers, PARP inhibitors, immunotherapy

## Abstract

Triple negative breast cancers (TNBCs) are characterized by worse prognosis, higher propensity to earlier metastases, and shorter survival after recurrence compared with other breast cancer subtypes. Anthracycline- and taxane-based chemotherapy is still the mainstay of treatment in early stages, although several escalation approaches have been evaluated to improve survival outcomes. The addition of platinum salts to standard neoadjuvant chemotherapy (NACT) remains controversial due to the lack of clear survival advantage, and the use of adjuvant capecitabine represents a valid treatment option in TNBC patients with residual disease after NACT. Recently, several clinical trials showed promising results through the use of poly ADP-ribose polymerase (PARP) inhibitors and by incorporating immunotherapy with chemotherapy, enriching treatment options beyond conventional cytotoxic agents. In this review, we provided an overview on the current standard of care and a comprehensive update of the recent advances in the management of early stage TNBC and focused on the latest emerging biomarkers and their clinical application to select the best therapeutic strategy in this hard-to-treat population.

## 1. Introduction

Triple negative breast cancer (TNBC) is clinically defined by the lack of the estrogen receptor (ER), progesterone receptor (PR), and human epidermal growth factor receptor 2 (HER2) expression. TNBC represents approximately 15–20 percent of breast cancers (BC) diagnosed worldwide and occurs more often in younger, black women and patients harboring mutations in breast related cancer antigens 1 and 2 (BRCA1/2). TNBC is a heterogeneous disease and, during the last decades, several classifications have been proposed according to specific histological and molecular characteristics of tumor [1]. High grade ductal invasive carcinoma is the most frequent histological type, even if special subtypes as well as medullary-like, apocrine, adenoid-cystic, and metaplastic are generally triple negative cancers.

TNBC is associated with worse prognosis with a high propensity to distant metastases (usually within two or three years from diagnosis) and shorter survival after recurrence. The pattern of relapses is characterized by a preferential metastatic spread to lungs, liver, and brain, while skeletal involvement is less common compared to luminal cancers [2]. Despite the emerging role of targeted therapies, TNBC is still an orphan disease in terms of therapeutic options, and chemotherapy remains the mainstay of treatment. Unfortunately, only about one third of patients responds to chemotherapy, thus the identification of actionable molecular drivers is crucial for the development of novel targeted treatments. Recently, several clinical trials showed promising results through the use of poly ADP-ribose polymerase (PARP) inhibitors and by incorporating immunotherapy with chemotherapy, enriching treatment strategies beyond conventional cytotoxic agents.

In this review, we provide an overview on the current therapeutic options and a comprehensive update of the latest advances in the management of early stage TNBC.

## 2. Classifications of TNBC: Molecular Subtypes and Gene Signatures

Gene expression microarray studies identified a spectrum of distinct molecular subtypes of BC with different prognosis and specific biological characteristics (Table 1). Based on gene expression array, Perou et al. proposed the first molecular classification, identifying four intrinsic subclasses of breast tumors: luminal A, luminal B, HER2-enriched, and basal-like subtype. These molecular portraits are defined according to the different clusters of co-expressed genes: hormone receptor (luminal cluster), Her2, cell-cycle regulating genes, and other genes mapping to chromosome 17 [3].

Immunohistochemistry and gene expression arrays demonstrated about 70–80% of concordance between triple negative disease and basal-like (BL) subtype. Approximately 20–25% of triple negative breast cancers (TNBCs) are not basal-like at microarray analysis, while 25% of basal-like tumors are not immunohistochemically triple-negative.

Additional novel molecular entities that preferentially display triple negative phenotype are the claudin-low and interferon-rich subtype. The claudin-low tumors are characterized by the absence of luminal differentiation markers, enrichment for epithelial-mesenchymal-transition, and immune response genes, low proliferation, cancer stem cell-like features, and poor prognosis [4], while the definition of interferon-rich subtype is based on the overexpression of interferon-regulated genes cluster such as STAT 1, associated with lymph node metastasis and worse outcome [5].

The advances in transcriptomic field have led to the identification of several molecular drivers that define seven clusters within TNBCs: basal-like 1 (BL1), basal-like 2 (BL2), mesenchymal- like (M), mesenchymal stem-like (MSL), immunomodulatory (IM), luminal androgen receptor (LAR), and unstable clusters. BL1 is characterized by the expression of basal cluster of genes involved in DNA repair mechanisms and cell-cycle regulation. BL2 shares with BL1 a highly proliferative phenotype, and it shows a marked overexpression of growth factor signaling genes and myoepithelial differentiation markers. M and MSL subtypes, which mainly belong to the claudin low intrinsic subgroup, are enriched for genes encoding regulators of cell motility, invasion, and mesenchymal differentiation. Furthermore, MSL shares with the IM type numerous genes engaged in the regulation of immune response, antigen processing and presentation, immune cell, and cytokine signaling pathways. Tumors with LAR subtype typically express high levels of androgen receptor (AR) and present gene alterations of phosphatidyl inositol 3-kinase (PI3K) pathway, such as PI3KCA (55%), AKT1 (13%), NF1 (13%), and CDH1 (13%) mutations, in conjunction with a higher prevalence of invasive lobular histology [6]. Furthermore, Lehmann and colleagues observed that the presence of distinctive stromal cells in tumor specimen was associated with specific behavior across subtypes in terms of both presentation and outcomes. This assumption led to a new classification of TNBC in four subgroups: BL1 (immune-activated), BL2 (immune-suppressed), M (including most of the MSL), and LAR [8]. Interestingly, the response to neoadjuvant chemotherapy (NACT) is largely influenced by the specific class of TNBC with the highest and the lowest pathological complete response (pCR) rates reported in BL1 (65.6%) and LAR (21.4%) subtypes, respectively [9]. Moreover, Burnstein and colleagues proposed a similar classification founded on a combination of RNA and DNA profiling analyses dividing TNBC into four subgroups: LAR, mesenchymal (MES), basal-like immunosuppressed (BLIS), and basal-like immune-activated (BLIA). Each subtype has well-defined molecular targets and different prognosis. Noteworthy, the BLIS subgroup exhibited the worst prognosis, while the BLIA subtype is related to favorable outcomes in terms of disease-free survival (DFS) [7].

Despite efforts in classifying TNBC, discordance in the number of molecular subtypes and methods used as well as intratumor genomic heterogeneity represented barriers in developing targeted therapies. However, by reviewing all genomic studies, it emerged that TNBC consists of four main subgroups: BL, mesenchymal, LAR, and immune-enriched. This simplified molecular portrait of TNBC provided the basis for a new subtype-driven therapeutic approach. For example, in BL cancers, targeting DNA damage response pathways by using platinum salts and PARP inhibitors (PARPis) could be an effective therapeutic strategy, while the mesenchymal subtype would seem to be more sensitive to PI3K, vascular endothelial growth factor 2 (VEGFR2), fibroblast growth factor receptor (FGFR), or mammalian target of rapamycin (mTOR) inhibitors [10]. Several phase I–II trials have suggested the activity of anti-androgen agents in the LAR subgroup, even if results from confirmatory phase III studies are needed to validate the efficacy of this strategy [11]. Furthermore, ongoing trials are investigating the usefulness of immunotherapy for treatment of patients affected by immune-enriched TNBC.

In conclusion, the implementation of next-generation sequencing (NGS) and other genomic assays into everyday practice is expected soon to deliver personalized treatment for TNBC, but, until then, patient enrollment in clinical trials that stratify according to the presence of a specific gene signature should be strongly encouraged.

## 3. BRCA1/2 Mutations

*BRCA1 and BRCA2* are tumor suppressor genes that exhibit an autosomal dominant pattern of inheritance with high penetrance. In the presence of DNA double-strand breaks (DSBs), the proteins encoded by these genes are involved in a conservative form of DNA-repair processes defined as homologous recombination repair (HRR) able to recover the original DNA sequence.

Many BRCA1/2 dysfunctions arise from germline mutations, promoter methylation, and somatic alterations. Patients with deleterious BRCA1/2 mutations are more sensible to alkylating agents, platinum salts, or PARPis, since these drugs induce irreparable DNA damage in hormone receptor (HR) deficient cells and consequently lead to cell cycle arrest and apoptosis [12]. Germline BRCA1/2 (gBRCA1/2) mutations are responsible for 52% and 32% of all hereditary breast cancers (BCs), respectively [13].

TNBC phenotype accounts for 71% of gBRCA1 mutations carriers while only 25% of patients with gBRCA2 mutations are affected by TNBC [14]. Noteworthy, gBRCA1/2 mutations have been identified in 10–20% of TNBCs, while somatic mutations are rarely reported (3–5% of cases) [15]. BRCA1 mutated TNBC patients are commonly younger than those harboring BRCA2 mutations, with a median age at diagnosis of 47.2 years and 58.8 years, respectively [16]. The relationship between these genomic scars and race/ethnicity has been widely studied, showing the lowest and the highest prevalence of gBRCA1/2 mutations in the Asian group (0.5%) and in the Ashkenazi Jewish (AJ) population (10.2%), respectively [17]. Interestingly, a recent analysis conducted in the USA showed that the incidence of pathogenetic BRCA2 mutations is higher in the AJ population compared to non-Hispanic whites, while BRCA1 alterations were not affected by race and ethnicity [18].

Several trials demonstrated the effectiveness of platinum-based chemotherapy for TNBC patients with BRCA mutations both in preoperative [19] and in metastatic settings [20]. However, two randomized clinical studies showed that the addition of platinum to standard NACT significantly increased pCR rate in TNBC regardless of the presence of gBRCA1/2 mutations [21]. Noteworthy, results from ongoing (neo)-adjuvant trials are awaited to clarify if PARPis could have a role in early stage BC as in advanced disease [22,23].

Almost 20% of BC patients share histological features and clinical outcome to BRCA1/2 related cancers without detectable gBRCA1/2 mutations, a phenotype defined as *BRCAness*. The most common mechanisms that cause BRCAness status are somatic mutations, large deletions, and DNA hypermethylation of BRCA1/2, germline mutations in PALB2, and hypermethylation of RAD51C genes. Recently, three single nucleotide polymorphism array-based signatures of chromosomal instability identified three distinct types of genomic scars caused by deficient DNA repair system: number of telomeric allelic imbalances (NtAI) [24], large scale transition (LST) [25], and loss of heterozygosity (LOH) [26]. The Homologous Recombination Deficiency (HRD) score is derived by the sum of these three metrics and represents an indirect measure of tumor genomic instability. Myriad genetics and myChoice CDx are the most commonly used assays to evaluate this signature. HRD status was defined as positive if either BRCA1/2 mutation or a predefined high HRD score (≥42) were revealed or classified as negative if no BRCA1/2 mutations or a low HRD score (<42) were detected [27].

The ability of HRD status to identify TNBCs likely to respond to platinum salts monotherapy or in combination was investigated in a large analysis of three neoadjuvant phase II studies (PrECOG 0105, Cisplatin-1, and Cisplatin-2). The results of this study strongly suggested that TNBC patients with HRD positive status treated with platinum-based NACT achieved more favorable response rates both in terms of pCR and residual cancer burden (RCB) compared to negative ones [28,29,30]. Similarly, in a retrospective analysis of Geparsixto, HRD TNBC patients randomized in the carboplatin arm achieved doubled pCR rates versus the non-HRD subgroup (63% and 29%, respectively), albeit without significant advantage in DFS or overall survival (OS) [31]. All these studies proved a strong correlation between HRD status and responsiveness to platinum-based NACT among TNBC patients. Noteworthy, in contrast with these data, platinum demonstrated no additional benefit compared to docetaxel in BRCAness metastatic TNBC, suggesting that its activity is not epigenetically driven by HRD status [20].

Finally, an alternative assay called “HRDetect” was recently generated to accurately detect BRCA1/2 deficiency. This tool, through whole-genome sequencing, identified six critically mutational signatures associated with BRCA1/2 deficiency that could predict benefits to platinum salts or PARPis therapies [32]. However, prospective clinical trials with a control arm and larger retrospective metanalysis are needed to clarify the clinical utility of the HRD status assessment.

All international guidelines highlighted the importance to identify gBRCA1/2 mutations carriers by performing genetic counseling as early as possible during the course of disease in patients at risk of inherited disease, given the implication in decision-making process in terms of the type of surgery, radiotherapy, and systemic treatment either in neo-(adjuvant) and advanced settings [33].

## 4. Other Biomarkers in Immunotherapy Era

### 4.1. Tumor Infiltrating Lymphocytes (TILs)

Considerable evidence highlights the relationship between host immune system, tumor microenvironment, and response to anticancer therapy. BC has been historically considered as a non-immunogenic disease characterized by a relatively low mutation rate. However, TNBC displayed the highest mutation rate and a marked presence of tumor infiltrating lymphocytes (TILs). TILs consist of mononuclear immune cells that infiltrate tumor tissue composed prevalently of cytotoxic CD8+ lymphocytes, and, to a lesser extent, CD4+ T helper cells, T regulatory (Treg) cells, macrophages, mast cells, and plasma-cells. Tumors with more than 50–60% of TILs in their specimen area are defined as lymphocyte-predominant breast cancer (LPBC), identifying a subset of TNBC associated with better outcome. Different percentages of LPBC were reported in TNBC patients enrolled in (neo)-adjuvant studies, ranging from 4.4 to 28.3% [34].

Several adjuvant and neoadjuvant trials demonstrated that the presence of intra-tumoral (iTILs) and stromal (sTILs) TILs represents a reliable surrogate of the immune anti-tumor activity and a robust independent prognostic biomarker in BC [35]. Recently, a pooled analysis of nine clinical trials confirmed sTILs as independent prognostic factors in early-stage TNBC, showing a significant correlation between higher levels of sTILs and improved survival after adjuvant chemotherapy [36]. Noteworthy, sTILs may add important information to conventional clinicopathologic factors in order to select a subset of patients who could not derive benefits from chemotherapy. In a large series of more than 500 women affected by pathological stage I TNBC, the presence of sTILs level of at least 30% identified a cohort of patients with excellent prognosis (5-year OS of 98%) without adjuvant chemotherapy [37]. Moreover, the presence of high TILs (>60%) in residual disease after NACT seemed to be correlated with better outcome for TNBC patients in which post-neoadjuvant treatment did not offer additional survival gain [38,39].

The predictive role of TILs, measured in pre-treatment biopsies, was also explored in neoadjuvant studies. A metanalysis of six randomized trials performed by the German Breast Cancer Group confirmed a significant pCR improvement reported in high TILs (≥60%) TNBC patients compared to low TILs subgroup (50% vs. 31%, respectively) [40].

TILs count was also investigated as a predictive marker of the response to immune checkpoint inhibitors (ICI) alone or in combination with chemotherapy. Data from Keynote-086 reported that sTILs count was able to predict efficacy of pembrolizumab as a first line treatment in metastatic TNBC [41].

In a neoadjuvant setting, a recent analysis explored the predictive role of TILs level and programmed death ligand 1 (PD-L1) expression, reported as combined positive score (CPS), in patients treated with immunotherapy-based combination. In the phase Ib Keynote-173 trial, patients who responded to different NACT regimens plus pembrolizumab showed higher levels of pre-treatment and on-treatment TILs as well as pre-treatment CPS. In particular, high pre-treatment TILs were observed in 40% of patients achieving pCR and in only 10% of the non-responder cohort [42].

Despite the prognostic and the predictive roles of TILs having been established, the value of qualitative characterization of immune-infiltrate is still under investigation. Breast tumors containing elevated counts of CD8+ lymphocytes with features of tissue-resident memory T-cell differentiation or a high CD8+/FOXP3+ ratio are deeply associated with improved survival after NACT, while the prognostic significance of regulatory T lymphocytes remains controversial [43].

In conclusion, TILs count is considered a well-established prognostic biomarker in TNBC (level of evidence I), and its clinical routine use is recommended by the 2019 Saint Gallen Consensus. However, to date, TILs scoring should not be used to guide neo-(adjuvant) treatment choices [44].

### 4.2. Programmed Death Ligand 1 (PD-L1)

PD-L1, a transmembrane protein predominantly involved in negative regulation of the T-cell function, is expressed in both tumor and immune cells (myeloid cells, T Reg, Natural Killer cells, endothelial cells) in various types of cancers. PD-L1 expression is commonly found in LPBC, and it is related to distinctive features such as younger age, high grade tumors, non-luminal HER2-positive and TNBC clinicopathological surrogates, as well as basal-like and HER2-enriched molecular subtypes. Despite the aggressiveness usually conferred by these characteristics, cancers with upregulated PD-L1 showed better OS and increased sensitivity to chemotherapy, suggesting a reciprocal interplay between immune response and antitumor activity [45,46]. PD-L1 assessed on immune cells (IC) is expressed in about 40–65% of TNBC tested samples, while its expression is lower (about 10%) when measured on tumor cells (TC) [47].

In metastatic settings, the predictive value of PD-L1 positivity on IC (PD-L1 IC +) was validated in several clinical trials. In particular, the IMpassion 130 study demonstrated that the addition of atezolizumab (anti-PD-L1) to nab-paclitaxel as first line treatment was associated with a relevant OS improvement of 7 months in PD-L1 IC ≥ 1% population (25 vs. 18 months) [48].

However, data from Keynote-522 and NeoTRIPaPDL1 trials [49,50] investigating the efficacy of pembrolizumab or atezolizumab in combination with conventional chemotherapy as a neoadjuvant approach revealed that PD-L1 expression significantly increased pCR rates irrespective of ICI addition. Therefore, the use of PD-L1 expression as a predictive marker of ICI efficacy remains controversial in early BC settings.

Noteworthy, the standardization of the method to determine PD-L1 positivity in BC is still unclear. A post-hoc exploratory analysis of the IMpassion130 trial evaluated the analytical concordance of VENTANA SP142 (IC ≥ 1%) with two other immunohistochemistry assays, SP263 (IC ≥ 1%) and Dako PD-L1 IHC 22C3 (CPS ≥ 1). The overall percentage agreements of the SP142 method with Dako 22C3 and SP263 were 69% and 63%, respectively, whereas a positive percentage agreement of 98% for both assays was estimated. Thus, SP142+ patients represent a subgroup of a larger PD-L1 positive population defined by Dako 22C3 and SP263 assays [51].

Furthermore, the reproducibility of PD-L1 detection due to its spatial and temporal heterogeneity and the different relevance as predictive markers of TC and IC positivity remains undefined. In order to overcome these gaps, new biomarkers are currently under evaluation as potential surrogates of ICI efficacy [52].

### 4.3. Tumor Mutation Burden (TMB) and Microsatellite Instability (MSI) and Mismatch Repair Deficiency (MMRd)

Tumor mutation burden (TMB) is a measurement of the total number of non-synonymous mutations per coding area of a tumor genome, which is easily evaluated by NGS-based techniques. These mutations are responsible for increased production of misfolded proteins (neoantigens) within cells recognized as non-self by the immune system. In contrast to other tumors (melanoma, lung cancer, and colorectal cancer), in which several data support TMB as a predictive biomarker for ICI efficacy, regardless of PD-L1 expression, its clinical utility in BC is not well known due to the limited and controversial data available. In BCs, high TMB is found in only 3.1% of cases and is more frequently detected in TNBC compared to luminal cancers [53].

In a retrospective analysis, patients with high TMB and favorable immune subclass defined by “positive” immune infiltrate disposition displayed improved survival independently from tumor stage, molecular subtype, age, and schedule of treatment [54]. Conversely, in a comprehensive analysis including clinical and genomic data, no differences in survival emerged for BC patients with high TMB treated with ICI-based therapy [55]. 

Furthermore, in the subgroup analysis of the neoadjuvant GeparNuevo study, early TNBC patients with high TMB achieved the highest pCR rate compared to those with low TMB with or without the addition of durvalumab to conventional NACT [56]. However, the absence of standardized assay and cut off value to define high mutational load represents a limit to the use of TMB as a biomarker in clinical practice.

Microsatellites are short DNA sequences repeated in tandem along the human genome and are particularly susceptible to misalignments during DNA replication. When the mismatch repair system is impaired, a condition defined as mismatch repair deficiency (dMMR), tumors exhibit a hypermutator phenotype called microsatellite instability (MSI). Recently, the FDA approved pembrolizumab for any unresectable or metastatic solid tumors with high frequency MSI (H-MSI) or dMMR [57]. Nevertheless, H-MSI/dMMR is an extremely rare condition in BC ranging between 0–1% of cases (0.6% of TNBC patients) and, to date, its clinical use as a prognostic and a predictive biomarker deserves further investigations [58].

## 5. Treatment of Early Stage

The therapeutic decision-making process represents a crucial point for early TNBC due to its higher propensity to relapse rapidly, generally within 2–3 years from diagnosis, with limited treatment options available in the metastatic setting beyond conventional cytotoxic drugs.

Chemotherapy is still the mainstay of therapy with no statistically or clinically significant difference between pre- or post-surgery administration [59], and regimens based on anthracyclines and/or taxanes remain the standard of care [60]. The neoadjuvant approach is the preferred choice for BC patients with locally advanced or inoperable disease to allow breast-conserving surgery and better cosmetic results, but the main advantage of NACT is the ability to pre-emptively predict treatment efficacy, driving physicians to optimize systemic strategies before surgery or to evaluate further post-operative treatments in case of residual disease.

Despite the aggressiveness, TNBC generally achieves a higher pCR rate after chemotherapy compared to other BC subtypes [61,62,63]. Several studies showed a strong correlation between pCR (defined as no invasive or in situ disease in the breast or the lymph nodes at time of surgery) and better outcomes, leading researchers to consider pCR as a surrogate endpoint for DFS in TNBC patients [63]. However, the use of pCR as a surrogate of long-term survival benefits is still debated, since no statistically significant correlation with OS has been formally demonstrated.

Residual disease post-NACT is a well-established biomarker of recurrence risk used to select patients for adjuvant escalated therapy. The quantity of residual disease in the surgical specimen, also called residual cancer burden (RCB), is classified in RCB-0, I, II, and III according to the size and the cellularity of the tumor detected in the surgical specimen (RCB-0 is equivalent to pCR, and RCB-III stands for no response or progression). The prognostic value of RCB was confirmed by Symmans and colleagues in a prospective trial enrolling 820 patients treated with various chemotherapy regimens containing anthracyclines and taxanes. Particularly, in patients affected by triple-negative disease, a 10-year DFS rates resulted of 86%, 81%, 55%, and 23% for RCB-0, RCB-I, RCB-II, and RCB-III, respectively [64]. Recently, patients with RCB after NACT were selected as appropriate candidates for intensified post-operative chemotherapy.

The CREATE-X trial randomized 910 HER2-negative BC Asiatic patients with residual disease after NACT to receive either eight cycles of adjuvant capecitabine or no further treatment [65]. In the TNBC cohort that represented one-third of the study population, patients receiving capecitabine had higher rates of 5-year DFS (70% vs. 56%) and OS (79% vs. 70%) compared to the control arm. Notably, the addition of capecitabine in patients with no-TNBC phenotype resulted in a numerically but no clinically meaningful advantage in terms of survival gain (93% vs. 90%, respectively).

Indeed, in the preliminary results of a phase III GEICAM/CIBOMA trial, which enrolled 876 Caucasian women with early-stage TNBC treated with (neo)-adjuvant chemotherapy, the extension with adjuvant capecitabine versus placebo did not significantly improve 5-year DFS (80% vs. 77%) and OS (86.2% vs. 85.9%) in the whole study population, although in a pre-specified subgroup analysis, a significant survival benefit was reported among patients with no basal-like TNBC [66]. 

Of note, a meta-analysis of trials conducted in America, Europe, and Asia showed a significant improvement in DFS and OS by the addition of adjuvant capecitabine in early TNBC patients pretreated with anthracycline and taxane-based chemotherapy irrespective of study region [67].

Finally, a recent meta-analysis of 12 prospective randomized trials including more than 15,000 BC patients reported that only the TNBC subset derived significant survival benefit from adjuvant capecitabine, particularly when it was given in addition to systemic treatment rather than instead of another therapy [68].

However, the role of adjuvant capecitabine in TNBC patients who received platinum-based chemotherapy and/or ICI as part of a neoadjuvant strategy remains to be investigated. Moreover, the results of the ongoing phase III trial comparing capecitabine with platinum salts in TNBC with residual disease after NACT are awaited (NCT02445391).

In clinical practice, the addition of adjuvant capecitabine should be considered in TNBC patients, especially in those with residual disease after standard NACT.

### 5.1. Anthracyclines

In the last 30 years, anthracyclines have represented the backbone of (neo)-adjuvant therapy for early BC patients. However, several phase III trials and metanalyses reported a modest benefit from anthracycline and taxane-containing regimens compared to anthracycline-free adjuvant chemotherapy in patients with high-risk disease.

A joint analysis of NSABP B-49, USOR 06-090, and NSABP B-46-I/USOR 0713 phase III trials (ABC trials) reported an absolute gain of 2.5% in 4-year invasive DFS in favor of anthracycline-containing regimen (90.7% vs. 88.2%) but no difference in terms of OS between anthracycline-based and anthracycline-free treatment. In a subgroup analysis, patients with TNBC and more extensive lymph nodal involvement seemed to derive greater survival benefit from taxanes plus doxorubicin/cyclophospamide (TaxAC), although the statistical significance was not met [69].

Conversely, data from the phase III West German Study (WSG) Plan B trial, which evaluated the non-inferiority of TC (docetaxel plus cyclophosphamide) versus epirubicin plus cyclophosphamide (EC) followed by docetaxel in the same study population of the aforementioned ABC trials, showed similar 5-year DFS and OS in both treatment arms irrespective of HR status (DFS = 89.9% vs. 89.6%; OS = 94.7% vs. 94.5%, respectively) [70]. These discordant findings could be explained by the higher percentages of triple negative and node positive BC patients included in the ABC trials compared with WSG Plan B study.

Finally, a pooled analysis of the ABC trials, the WSG Plan B trial, and a study performed by the Hellenic Oncology Research group (HORG) failed to demonstrated the non-inferiority of TC regimen compared to TaxAC, although the absolute benefit in DFS was relatively limited (1.28%) [71].

Basing on these controversial results, and taking into account the long-term cardiotoxicity derived from anthracycline-containing chemotherapy, TC regimen is considered a valid alternative option for patients with very low-risk TNBCs (node-negative and T < 1 cm) or in case of severe cardiac comorbidities.

A robust body of evidence shows statistically significant survival benefit with the use of adjuvant dose-dense chemotherapy, in which anthracyclines are given at two weekly intervals compared to the standard three weekly administration in patients with HR negative tumors (HR 0.8, *p* = 0.002) but not in HR positive BC cohorts [72].

A more recent large meta-analysis of 26 trials of adjuvant therapy clearly showed that dose-dense chemotherapy improves the outcome in terms of BC recurrence (28% vs. 31.4%), BC specific mortality (18.9% vs. 21.3%), and overall mortality (22.1% vs. 24.8%) with similar safety profiles. However, the benefit was not affected by hormone receptor status [73].

In neoadjuvant settings, the role of the dose dense strategy is still debated, since no consistent advantages in pCR and long-term outcomes have been demonstrated [74]. However, based on the intrinsic aggressiveness and the higher proliferative rate of TNBC, the dose dense approach represents an attractive choice for neoadjuvant treatment, where an intensive schedule of administration could reduce the re-growth of cancer cells during the interval between cycles [75].

To date, dose dense neo-(adjuvant) chemotherapy represents the preferred schedule for high-risk BC patients.

### 5.2. Taxanes

Several trials and metanalyses suggested the activity of paclitaxel and docetaxel in adjuvant settings, particularly in high risk patients such as TNBC, HER2 positive tumors, and high grade or node positive luminal BC, showing a significant reduction in the risk of recurrence and mortality compared to taxane-free chemotherapy.

The CALGB 9344/INT1048 trial demonstrated firstly the efficacy of paclitaxel added to doxorubicin and cyclophosphamide AC in node positive BC patients. This study randomized 3121 patients to receive AC for four cycles followed by four cycles of paclitaxel or AC alone administered every three weeks. The addition of paclitaxel resulted in an absolute improvement in 5-year disease-free and overall survival of 5% and 3%, respectively. Noteworthy, among HER2 negative patients, an unplanned analysis reported that the beneficial effect of paclitaxel was confined to HR negative subgroup (HR = 0.72) [76].

In the GEICAM 9906 adjuvant trial, the subset of TNBC patients treated with fluorouracil, epirubicin, and cyclophosphamide (FEC) followed by paclitaxel derived an absolute benefit of 18% in DFS at 7 years follow up compared to FEC alone (DFS 74% vs. 56%, respectively) [77]. Similarly, in the French PACS 01 trial comparing six cycles of FEC to three cycles of FEC followed by three cycles of docetaxel, patients with BL subtype obtained the greater DFS benefit from the sequential chemotherapy regimen [78].

Two exploratory analyses performed in the E2197 study [79] and the CALGB trials [80] confirmed that taxanes were more effective in HR-negative tumors compared with HR-positive ones. However, an updated Cochrane systematic review including 29 adjuvant trials reported increased survival by adding taxanes to standard anthracycline-based chemotherapy with no different treatment effect by subgroups for HR status [81].

In the neoadjuvant context, the NSABP B-27 trial assigned 2411 women to receive either four cycles of standard AC every 3 weeks followed by surgery, or four cycles of AC followed by four cycles of docetaxel and then surgery, or four cycles of AC followed by surgery and then four cycles of adjuvant docetaxel (D), which displayed that the addition of preoperative docetaxel to AC (AC-D) improved response regardless of HR status, even if the pCR rate was nearly double in the HR negative subgroup compared to the HR positive population (22.8% vs. 14.1% for AC-D) [82]. Similarly, in the GeparDuo and the I-SPY neoadjuvant trials, pCR rates were significantly higher in the HR negative versus the HR positive subset [83,84]. Furthermore, a retrospective analysis of seven consecutive neo-adjuvant studies including 1079 patients who received NACT with or without taxanes showed that the pCR rate was enhanced in the HR negative subgroup treated with taxanes (pCR: 29% vs. 15%) [85].

Traditionally, docetaxel and paclitaxel have been considered equivalent and interchangeable. Sparano et al. evaluated the efficacy of AC followed by weekly (q1w) or three weekly (q3w) docetaxel or paclitaxel in the adjuvant setting including 4950 early BC patients. The results demonstrated an improvement in both DFS and 5-year OS with q1w paclitaxel versus q3w administration. In TNBCs, q1w paclitaxel compared to q3w regimens (docetaxel or paclitaxel) was associated with an approximately 30% reduction in the risk of recurrence and death and with an absolute gain in terms of DFS and OS of 10% at long term follow up [86].

Interestingly, in order to improve efficacy and decrease toxicities related to the use of conventional taxanes, in the last decade, novel formulation was developed. The neoadjuvant GeparSepto trial compared weekly nanoparticle albumin-bound paclitaxel (nab-paclitaxel) to conventional paclitaxel for 2 weeks, both followed by four cycles of EC, 1200 patients with HER2 positive and HER2 negative early BC. The improvement of pCR rate was mainly observed among the 275 patients affected by TNBC (48% with nab-paclitaxel versus 26% with standard paclitaxel). Noteworthy, a significant invasive DFS benefit was reported in the TNBC subgroup who achieved pCR, supporting the value of pCR as the surrogate endpoint for long-term outcomes in this population [87].

Conversely, in the phase III ETNA trial conducted in 695 patients with HER2 negative early or locally advanced BC, a slight and not statistically significant improvement in pCR rate was registered in the nab-paclitaxel arm compared to paclitaxel cohort (22% vs. 19%), even in patients with TNBC (41% vs. 37%) [88]. Likewise, the secondary endpoint of 5-year event free survival (EFS) was not statistically different between neoadjuvant nab-paclitaxel compared to standard paclitaxel [89].

The conflicting data between the two trials could be explained in part by the different populations of patients enrolled and the administration schedule of nab-paclitaxel used (3 weeks on/1 week off in ETNA vs. every week in the GeparSepto study).

Nonetheless, nab-paclitaxel is currently recommended only for high risk disease and for patients who have experienced serious adverse (hypersensitivity or anaphylaxis) from conventional taxanes.

### 5.3. Platinum-Based Chemotherapy

TNBCs are typically more sensitive to DNA-damaging compounds, such as platinum agents, due to the high prevalence of DNA repair pathways defects reported in these tumors.

Results from two large phase II trials showed a significant improvement in pCR rates for TNBC patients by the addition of carboplatin to neoadjuvant anthracycline and taxane-based chemotherapy. However, it is not clear whether this effect on pCR would ever translate into lower recurrence rate. In the Geparsixto trial, TNBC patients derived an absolute benefit in 3-year EFS of about 10% when carboplatin was added to standard NACT plus bevacizumab (85.8% vs. 76.1%, respectively), although no statistically significant advantage in OS was registered [90].

In contrast, the higher pCR rates obtained by the incorporation of carboplatin into conventional NACT led to nonsignificant gain in terms of 3-year EFS in the subgroup of patients with TNBC enrolled in the CALGB 40603 study [91].

The difference between these trials may arise from the inclusion of patients with more favorable baseline characteristics in the CALGB 40603 study compared with the Geparsixto population. Furthermore, in CALGB 40603, the use of cyclophosphamide combined with anthracycline, which also induces DNA damage like platinum salts, could potentially undermine the effect of carboplatin between the control and the experimental arms. Noteworthy, these studies were not powered to demonstrate survival differences.

Furthermore, the BrighTNess phase III trial, which evaluated the addition of the PARP inhibitor veliparib plus carboplatin or carboplatin alone to standard NACT in stages II–III TNBC patients, reported substantially higher pCR rates in patients receiving carboplatin with or without veliparib (53% and 58%, respectively) compared to the subgroup assigned to paclitaxel alone (31%) [92].

Interestingly, a recent metanalysis of nine randomized clinical trials demonstrated an absolute gain of 15% in pCR rates by the use of neoadjuvant platinum-based chemotherapy with the exception of the GEICAM/2006-03 study. The incorporation of cyclophosphamide in the standard NACT regimen of GEICAM/2006-03 study could partially explain the lack of benefit in pCR achieved, since the alkylating agent, acting as a DNA-damaging drug, may decrease the positive effect of platinum addition [21].

BRCA status has been studied as a predictive biomarker of response to platinum agents, although its role has not been yet established. GeparSixto and BrighTNess trials reported similar pCR rates in BRCA mutated cohorts irrespective of the addition of platinum (58.0% vs. 54.3% in the platinum-based chemotherapy group vs. the platinum-free chemotherapy group). However, the effectiveness of platinum agents in neoadjuvant setting remains an unsolved question, since these data were derived from a post-hoc exploratory analysis with limited number of BRCA mutated patients (95/609 patients) [21]. Results from the ongoing phase II neoadjuvant INFORM trial are awaited to further define the utility of platinum compounds in BRCA mutated settings.

## 6. Novel Therapeutic Options and Future Perspectives

### 6.1. PARP Inhibitors

On the basis of positive results of the phase III studies, OlympiAD and EMBRACA [22,23], olaparib and talazoparib were recently approved for treatment of metastatic HER2 negative BC patients harboring gBRCA1/2 mutation. As a consequence, several trials are evaluating the activity of PARPis in early settings to reduce toxicity and improve pCR rates and long-term outcomes compared to standard chemotherapy. Veliparib was the first PARP inhibitor used in the neoadjuvant phase II I-SPY2 trial, in which the addition of veliparib and carboplatin to standard NACT improved the probability of pCR (estimated by Bayesian analysis) compared to the control arm in TNBC patients (52% vs. 26%) but not in HR positive ones [93]. Surprisingly, the phase III Brightness trial, which enrolled 634 operable TNBC patients, failed to demonstrate a significant benefit in terms of pCR from the addition of veliparib to neoadjuvant carboplatin plus paclitaxel [92]. Furthermore, no difference in pCR emerged among gBRCA1/2 mutation carriers in an unplanned subgroup analysis [92]. The results of these two trials suggested that the observed improvement in pCR was mainly derived from the incorporation of carboplatin to standard NACT rather than to the addition of PARPis.

Talazoparib, the PARP inhibitor with the highest catalytic activity and the most efficient trapping mechanism, in a single arm phase II trial including patients with operable gBRCA1/2 mutated BC, achieved a pCR rate of 53% when administered in monotherapy for six months before surgery followed by standard adjuvant treatment [94]. Notably, the majority of patients enrolled had TNBC (15/20) and, among them, almost 50% achieved pCR.

Olaparib, the first PARP inhibitor approved for the treatment of metastatic BC harboring gBRCA1/2 mutation, is currently under investigation in phase II/III trials in early settings. The phase II GeparOla study randomized 102 operable HRD BC patients to receive either paclitaxel plus olaparib or paclitaxel plus carboplatin followed by EC as a neoadjuvant treatment. The preliminary results reported that TNBC patients derived no benefit in pCR from the addition of olaparib to conventional NACT [95].

Finally, rucaparib in combination with cisplatin was compared to cisplatin alone in TNBC patients with residual disease after NACT showing no significant contribution in terms of DFS derived from PARP inhibitor addition [96]. Although talazoparib showed encouraging results, several issues remain debated, such as the optimal dose and the identification of predictive biomarkers of response to PARPis beyond BRCA1/2 mutations. Confirmatory data from larger phase II/III randomized controlled trials are awaited (Table 2).

### 6.2. Immunotherapy for Early TNBC

BC is generally considered a “non-immunogenic” tumor due to its low mutational load and poor immune infiltrate. However, the TNBC subtype displays several features, such as the presence of TILs, the expression of immune evasion molecules in a tumor microenvironment such as PD-L1, and the genomic instability and consequently the highest number of mutations that make it a “hot” and immunogenic tumor.

Over the last years, several clinical studies showed that immunotherapeutic agents alone or in combination with standard chemotherapy yielded an increased overall response rate and survival in metastatic TNBC patients [48], supporting ICI as a valid approach in early BC settings.

The I-SPY 2 phase II trial evaluated the addition of pembrolizumab to standard NACT, showing a 40% improvement in the probability of pCR compared to the control arm (60% vs. 22%) [97]. Furthermore, in the phase Ib Keynote-173 trial, a multicohort study assessing the safety and the efficacy of pembrolizumab combined with various schedules and doses of platinum and taxanes followed by AC as NACT, the pCR rate across all cohorts ranged from 60% to 80%, with the best response registered in the subgroup of patients treated with pembrolizumab added to nab-paclitaxel plus carboplatin [98].

In the Keynote-522, the first phase III trial that demonstrated the effectiveness of anti-PD1 therapy in neoadjuvant setting, 1174 TNBC patients were randomized 2:1 to receive chemotherapy (carboplatin + paclitaxel followed by AC or EC) plus pembrolizumab or placebo followed by pembrolizumab alone or placebo after surgery. In this study, the addition of the pembrolizumab resulted in a statistically significant and clinically meaningful increase in pCR rate of 13.6% (64.8% vs. 51.2%, *p* = 0.00055) regardless of the PD-L1 expression status [49]. Data from the subgroup analysis presented at the last San Antonio Breast Cancer Symposium (SABCS) showed that patients with the most aggressive disease, such as stage III tumors, derived more benefit from the incorporation of the anti-PD1 to chemotherapy. Moreover, a favorable trend for EFS was revealed, although statistical significance was not already reached [99].

However, conflicting data arose from the phase II GeparNuevo and the phase III NeoTRIPaPDL1 trials [50,100]. The GeparNuevo, in which 174 patients were randomized to receive nab-paclitaxel plus durvalumab (anti PD-L1) or placebo followed by EC, failed to demonstrate a significant difference in terms of pCR between the two arms. Interestingly, in an unplanned analysis, the cohort of patients who received additional induction therapy with durvalumab before surgery achieved a better pCR rate (61.0% vs. 41.4%), suggesting that immunotherapy could enhance the anti-tumor activity of cytotoxic drugs [100].

The NeoTRIPaPDL1 study investigated the efficacy of atezolizumab in combination with carboplatin plus nab-paclitaxel in the neoadjuvant treatment of early high risk or locally advanced TNBC followed by AC or EC after surgery. The preliminary results showed that the combination therapy did not significantly improve the pCR rate compared to chemotherapy alone (43.5% vs. 40.8%, *p* = 0.66), even if a slight benefit was reported in PD-L1 positive patients assigned to the experimental arm (51.9% vs. 48%) [50].

The discordant results derived from Keynote-522 [49] and NeoTRIPaPDL1 [50] phase III studies could be explained by different ICI tested but also by the different chemotherapy regimens used. Indeed, in the Keynote-522, anthracyclines administered before surgery may have acted synergistically with immunotherapy due to their ability to induce immunogenic cell death and increase proliferation of CD8+ lymphocytes. Furthermore, since pCR is merely considered a surrogate endpoint of survival outcomes, long-term results of the NeoTRIPaPDL1 trial are awaited to definitively establish the effectiveness of ICI treatment in terms of EFS.

Finally, several ongoing trials are testing the ability of ICI monotherapy or in addition to conventional cytotoxic drugs to improve the “cure rate” in adjuvant setting (Table 3). 

### 6.3. Potential Therapeutic Targets in Early TNBC

The vascular endothelial growth factor (VEGF) pathway plays a key role in the pathophysiology of TNBC, as demonstrated by elevated vascular levels of VEGF and higher microvascular density compared to other subtypes [101].

In three phase III neoadjuvant trials (GeparQUINTO, ARTemis, and CALGB 40603), the addition of bevacizumab to anthracycline and taxane-based regimen increased pCR rate compared to chemotherapy alone without survival benefit [91,102,103]. In two large, randomized studies, BEATRICE (Bevacizumab Adjuvant Therapy in Triple-Negative Breast Cancer) and E5103, the combination of bevacizumab with standard adjuvant chemotherapy failed to show a favorable effect in terms of DFS in TNBC patients [104,105].

The epidermal growth factor receptor (EGFR) is frequently over-expressed in TNBC, and it seems related to a poor prognosis [106]. Anti-EGFR targeted therapies, such as tyrosine kinase inhibitors and monoclonal antibodies used in monotherapy or in combination with chemotherapy, were also evaluated for treatment of TNBC patients. To date, two phase II trials evaluated the incorporation of panitumumab or cetuximab to NACT in operable TNBC reporting a slight improvement in pCR. Interestingly, in both trials, TILs seemed to predict response to neoadjuvant anti-EGFR treatment [107,108]. Several clinical studies evaluating the effectiveness of anti-EGFR agents are currently ongoing in early settings.

The androgen receptor targeting pathway may represent a promising strategy for treatment of metastatic TNBC expressing AR on immunochemistry [109,110], and a neoadjuvant trial is currently ongoing to evaluate the combination of enzalutamide plus paclitaxel in patients with stages I–III AR positive TNBC (NCT02689427).

The PI3K-Akt-mTor pathway represents an attractive therapeutic target, since PIK3CA hotspots mutations and PTEN aberrations are frequently reported in TNBC [111]. However, a phase II study investigating the efficacy of the mTOR inhibitor everolimus in addition to standard NACT reported disappointing results [112].

Mitogen activated protein kinase pathway and FGFR signaling are often deregulated in TNBC and are generally related to chemo-resistance. Therefore, a great deal of effort was spent to evaluate MEK and FGFR targeting inhibitors for treatment of TNBC patients but, unfortunately, no data in early settings are available [113,114].

The promising results derived from ICI based therapy encouraged the researchers to evaluate novel immunomodulatory agents with distinct mechanisms of action such as anticytotoxic T lymphocyte-associated antigen 4 (CTLA-4), monoclonal antibodies (ipilimumab, tramelimumab, enoblituzumab), indoleamine 2,3 dioxygenase (IDO) inhibitor (epacadostat), and adenosine A2A receptor antagonist (CPI-444) with the aim of enhancing the anti-tumor immune response beyond anti-PD1/PD-L1 drugs [115].

The TNBC microenvironment is characterized by pro-inflammatory cytokines and chemokines, and an alternative immunotherapy strategy based on the silencing of cytokine receptors represents an intriguing field of research [116]. In this context, the deregulation of multiple copies in T-cell malignancy 1/miR-34a/interleukin-6/interleukin-6 receptor (MCT-1/miR-34a/IL-6/IL-6R) signaling axis, involved in epithelial–mesenchymal transition, M2-like macrophages polarization, and TNBC cell invasiveness, was identified as a potential therapeutic target. Pre-clinical trials showed interesting results by the use of MCT-1 inhibitor in combination with anti-IL-6/IL-6R monoclonal antibody, tocilizumab, or with short/small hairpin RNA, thus allowing further investigations [117].

Moreover, a synergistic effect by the addition of targeted therapies (PI3K, MEK, VEGF, EGFR, and PARPis) that are able to reinforce immune microenvironments to ICI is expected and currently under evaluation in ongoing clinical trials. Preliminary positive data are derived from trials exploring novel immune system modulators in TNBCs as well as adaptive cell therapy, chimeric antigen receptor (CAR) T cell therapy, and vaccines [118].

Finally, in the transcriptomic era, the identification of multiple potentially actionable targets led to categorizing TNBC into smaller and smaller subgroups. In this scenario, several biomarker-driven neo-adjuvant trials are investigating the efficacy of targeted therapies stratifying patients before randomization according to distinct gene signature with the aim to select a specific population to include in confirmatory phase III studies. A list of the ongoing phase II trials is reported in Table 4.

## 7. Conclusions

TNBC are associated with worse prognosis, higher propensity to earlier metastases, and shorter survival after recurrence compared with other BC subtypes. Standard chemotherapy with anthracyclines and taxanes is still the mainstay of systemic treatment in early settings.

The addition of platinum-salts to standard NACT significantly improved pCR rate in stages II–III TNBC patients, but its use is not uniformly endorsed by current guidelines due to the lack of clear survival benefit. Furthermore, in patients with residual disease after NACT, an escalation approach with post-operative capecitabine represents an effective option in prolonging survival.

The emerging cancer genomic data led to the development of new therapeutic strategies targeting specific molecular drivers. PARPis showed encouraging results for the treatment of TNBC, especially in patients with impaired DNA damage repair system. However, their use remains restricted to gBRCA1/2 metastatic BC disease.

Despite the impressive results of immunotherapy in advanced stage TNBC, the incorporation of ICI agents deserves further evaluation to validate their efficacy in (neo)-adjuvant settings.

Finally, the identification of reliable predictive biomarkers is imperative to offer the best therapeutic option for this hard-to-treat population.

## Figures and Tables

**Table 1 cancers-12-00819-t001:** Classifications of triple negative breast cancer (TNBC).

Subtypes	Characteristics
LEHMANN CLASSIFICATION [6]	
Basal-like 1	High level of DNA repair proteins and cell-cycle regulation (high rate of tp53 mutations, amplification of MYC, CDK6, CCNE1, deletion of BRCA-2, PTEN, MDM2, RB1).
Basal-like 2	Overexpression of growth factor signaling genes and overexpression of myoepithelial differentiation genes.
Mesenchymal like	Expression of genes associated with EMT.
Mesenchymal-stem like	Expression of genes associated with EMT; Enrichment in genes involved in angiogenesis, including VEGFR2 and some components of immune signaling; High expression of stem cells genes; Low expression of proliferation genes and epithelial-related genes involved in the maintenance of cellular junction, such as claudin (claudin-low breast cancer).
Immunomodulatory	Enrichment in genes involved in regulation of immune response, antigen processing and presentation, immune cells and cytokine signaling pathways.
Luminal androgen receptor (LAR)	High level of androgen receptor genes; Alterations of PI3K pathway genes (PI3KCA, AKT1, NF1, CDH1).
BURSTEIN CLASSIFICATION [7]	
Basal-like immunosuppressed (BLIS)	Downregulation of B cell, T cell, and natural killer cell immune-regulating pathways, and cytokine pathways; Expression of multiple SOX family transcription factors.
Basal-like immunoactivated (BLIA)	Upregulation of genes involved in immune cell function regulation; High expression of STAT genes.
Mesenchymal (MES)	Expression of genes involved in cell cycle, mismatch repair, and DNA damage networks, and hereditary breast cancer signaling pathways; Expression of genes normally exclusive to osteocytes (OGN) and adipocytes (ADIPOQ, PLIN1) and important growth factors (IGF1).
Luminal androgen receptor (LAR)	Expression of AR, ER, prolactin, and ErbB4 signaling genes.
MICROARRAY-BASED CLASSIFICATION [4,5]	
Basal-like	Low expression of luminal A signature, high proliferation score, low expression of estrogen signaling related genes (FOXA1, PGR); High expression of cell-cycle related genes (CCNE, FANCA); High expression of EMT (TWIST1, ZEB1).
Claudin-low	Low levels of cell adhesion proteins (Claudins 3, 4, 7, Occludin, E-caderin); Low expression of luminal genes; Inconsistent basal genes expression; Elevated expression of immune-related genes (CD4, CD79a).
Molecular apocrine	Activation of AR pathways.
Interferon-rich	Overexpression of interferon-regulated genes (STAT1, SP110).

Epithelial-mesenchymal transition, EMT; androgen receptor, AR; estrogen receptor, ER.

**Table 2 cancers-12-00819-t002:** Ongoing (neo)-adjuvant phase II-III trials with PARP inhibitors.

Phase	NCT	Study Population	Setting	Stage	Experimental Arm	Control Arm	Primary Endpoint
II	NCT03499353 (NeoTala)	TNBC and/or gBRCA1/2 mutated BC	Neoadjuvant	I–III	Talazoparib	NA	pCR
II	NCT02282345	gBRCA1/2 mutated BC	Neoadjuvant	I–III	Talazoparib	NA	iDFS
II	NCT02789332 (GeparOla)	HER2 negative BC with gBRCA1/2 mutation and/or HRD	Neoadjuvant	I–III	Standard NACT + Olaparib	Standard NACT + Carboplatin AUC2	pCR
II	NCT01042379	TNBC	Neoadjuvant	II–III	Veliparib + carboplatin → standard NACT	Standard NACT	pCR
III	NCT02032277 (BrighTNess)	TNBC	Neoadjuvant	II–III	Veliparib + carboplatin + paclitaxel → AC	Placebo ± carboplatin+ paclitaxel → AC	pCR
II/III	NCT03150576 (PARTNER)	TNBC and/or gBRCA mutated BC	Neoadjuvant	II–III	Olaparib + carboplatin + paclitaxel → AC/EC	Paclitaxel + carboplatin → AC/EC	Safety, pCR
III	NCT02032823 (Olympia)	HER2 negative BC with gBRCA1/2 mutation	Adjuvant	NA	Olaparib maintenance up to maximum 1 year	Placebo	iDFS

Doxorubicin plus cyclophosphamide, AC; epirubicin plus cyclophosphamide, EC; homologous recombination deficiency, HRD; pathological complete response, pCR; invasive disease-free survival, iDFS. not available, NA.

**Table 3 cancers-12-00819-t003:** Ongoing (neo)-adjuvant phases II–III clinical trials with immune checkpoint inhibitors.

Phase	NCT	Study Population	Setting	Stage	Experimental Arm	Control Arm	Primary Endpoint
II	NCT03639948 (NeoPACT)	TNBC	Neoadjuvant	I–III	Carboplatin + docetaxel + pembrolizumab	NA	pCR
II	NCT03289819	TNBC	Neoadjuvant	I–III	Pembrolizumab + nab-paclitaxel → pembrolizumab + EC	NA	pCR
II	NCT03356860 (B-IMMUNE)	TNBC and Luminal B	Neoadjuvant	I–III	Paclitaxel → EC + durvalumab	Paclitaxel + epirubicin → cyclophosphamide	Safety, pCR
II	NCT02685059 (GeparNuevo)	TNBC	Neoadjuvant	I–III	Durvalumab + nab-paclitaxel → EC	Placebo + nab-paclitaxel → EC	pCR
III	NCT02620280 (NeoTRIPaPDL1)	High-risk TNBC	(Neo)-adjuvant	II–III	Carboplatin + nab-paclitaxel + atezolizumab → AC/EC/FEC	Carboplatin + nab-paclitaxel → AC/EC/FEC	EFS
III	NCT03036488 (Keynote-522)	TNBC	(Neo)-adjuvant	II–III	Carboplatin + paclitaxel + pembrolizumab → AC/EC + pembrolizumab	Carboplatin + paclitaxel + placebo → AC/EC + placebo	pCR, EFS
III	NCT03281954 (NSABP B-59)	TNBC	(Neo)-adjuvant	II–III	Paclitaxel + carboplatin + atezolizumab → atezolizumab + AC/EC	Paclitaxel + carboplatin + placebo → placebo + AC/EC	pCR, EFS
III	NCT03197935 (IMpassion031)	TNBC	(Neo)-adjuvant	II–III	Nab-paclitaxel + atezolizumab → AC + atezolizumab → atezolizumab	Nab-paclitaxel + placebo → AC + placebo → placebo	pCR
III	NCT02954874 (SWOG 1418)	TNBC or ER and PgR ≤ 5% with residual disease ≥ 1 cm and/or ypN+	Adjuvant	-	Pembrolizumab	NA	iDFS, Safety
II	NCT03756298	TNBC with residual disease ≥ 1cm and/or ypN+	Adjuvant	-	Capecitabine + atezolizumab	Capecitabine	iDFS
III	NCT03498716 (IMpassion030)	TNBC and PD-L1+	Adjuvant	II–III	Paclitaxel + atezolizumab → dose-dense AC/EC	Paclitaxel → dose-dense AC/EC	iDFS
III	NCT02926196 (A-Brave)	High risk TNBC	Adjuvant	-	Avelumab	NA	DFS
II	NCT03872505 (PANDoRA)	TNBC	Neoadjuvant	II–III	Durvalumab + carboplatin + paclitaxel + radiation	Durvalumab + carboplatin + paclitaxel	pCR
II	NCT03546686	TNBC	(Neo)-adjuvant	I–III	Ipilimumab + nivolumab + cryoablation + breast surgery → nivolumab	Breast surgery	EFS

Doxorubicin plus cyclophosphamide, AC; epirubicin plus cyclophosphamide, EC; pathological complete response, pCR; event free survival, EFS; invasive disease-free survival, iDFS.

**Table 4 cancers-12-00819-t004:** Ongoing (neo)-adjuvant clinical trials with potential therapeutic targets.

Phase	NCT	Study Population	Setting	Stage	Experimental Arm	Control Arm	Primary Endpoint
II	NCT03348098	TNBC	Neoadjuvant	II–III	Apatinib + paclitaxel	NA	ORR
II	NCT03650738	TNBC	Neoadjuvant	II–III	Apatinib + nab-paclitaxel + carboplatin	NA	pCR, safety
II	NCT02511847	TNBC	Neoadjuvant	II–III	Afatinib + weekly paclitaxel	NA	pCR
II	NCT02720185	EGFR positive TNBC	Neoadjuvant	I–III	Dasatinib	NA	Increase in plasma membrane EGFR expression
II	NCT02750358	AR positive TNBC	Adjuvant	I–II–III	Enzalutamide	NA	Feasibility
NA	NCT03756090	TNBC	Neoadjuvant	I–III	Palbociclib + dose dense EC + paclitaxel	Placebo + dose dense EC + paclitaxel	pCR

Objective response rate, ORR; epidermal growth factor receptor, EGFR; androgen receptor, AR; pathological complete response, pCR; epirubicin plus cyclophosphamide, EC.

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
