# Peer review of "Early Triple Negative Breast Cancer: Conventional Treatment and Emerging Therapeutic Landscapes"

_cancers, 2020, doi:10.3390/cancers12040819_

Round 1

Reviewer 1 Report

Review of this manuscript "Early Triple Negative Breast Cancer: conventional treatment and emerging therapeutic landscapes" by Anna et. Al. The authors have reviewed the conventional or currently experimental treatments and  effectiveness for Triple Negative Breast Cancer (TNBC).   The manuscript is well organized. The current classification and clinical treatment of TNBC genotypes are introduced. I have some suggestions as follows: 1. Because BRCA1 / 2 mutation is racially diverse, I have not seen the authors discuss the effects on TNBC either in terms of treatment or cause.   2. Regarding immunotherapy, it is not enough to discuss PD-1 and PDL-1. More treatments and effects on immunosuppression should be discussed (such as MCT-1 oncogene and IL-6 / IL-6R immunotherapy).

Author Response

Dear Reviewer, we would like to thank you for careful and thorough reading of our manuscript titled “Early Triple Negative Breast Cancer: conventional treatment and emerging therapeutic landscapes”. We are sure that your suggestions will make more informative this review article.

Response to Reviewer 1 comments:

The manuscript is well organized. The current classification and clinical treatment of TNBC genotypes are introduced. I have some suggestions as follows:

Point 1. Because BRCA1 / 2 mutation is racially diverse, I have not seen the authors discuss the effects on TNBC either in terms of treatment or cause.  

Response 1: We added the following sentences discussing about the correlation between TNBC and BRCA1/2 carrier status and focusing on the relationship between the presence of BRCA1/2 mutations and race/ethnicity. Furthermore, we highlighted the importance of BRCA1/2 mutation carriers’ detection as it affects the therapeutic decision-making process.

Page 5, lines 123-124. “TNBC phenotype accounts for 71% of gBRCA1 mutations carriers while only 25% of patients with gBRCA2 mutations is affected by TNBC [14]”.

Page 5, lines 125-132. “BRCA1 mutated TNBC patients are commonly younger than those harbouring BRCA2 mutations, with a median age at diagnosis of 47.2 years and 58.8 years, respectively [16]. The relationship between these genomic scars and race/ethnicity has been widely studied, showing the lowest and highest prevalence of gBRCA1/2 mutations in the Asian group (0.5%) and in the Ashkenazi Jewish (AJ) population (10.2%), respectively [17]. Interestingly, a recent analysis conducted in USA showed that the incidence of pathogenetic BRCA2 mutations is higher in AJ population compared to non-Hispanic whites while BRCA1 alterations were not affected by race and ethnicity [18]”.

Page 6, lines 169-172. “All international guidelines highlighted the importance to identify gBRCA1/2 mutations carriers by performing genetic counseling as early as possible during the course of disease in patients at risk of inherited disease, given the implication in decision making process in terms of the type of surgery, radiotherapy and systemic treatment either in neo-(adjuvant) and advanced settings [33]”.

Point 2. Regarding immunotherapy, it is not enough to discuss PD-1 and PDL-1. More treatments and effects on immunosuppression should be discussed (such as MCT-1 oncogene and IL-6 / IL-6R immunotherapy).

Response 2:  We discussed about more potential treatment options with distinct mechanisms of action, beyond PD-1/PD-L1. We also mentioned MCT-1 signaling axis as requested.

Pages 17-18, lines 578-597. “The promising results derived from ICI based therapy encouraged the researchers to evaluate novel immunomodulatory agents with distinct mechanisms of action such as anticytotoxic T lymphocyte-associated antigen 4 (CTLA-4), monoclonal antibodies (ipilimumab, tramelimumab, enoblituzumab), indoleamine 2,3 dioxygenase (IDO) inhibitor (epacadostat), adenosine A2A receptor antagonist (CPI-444), with the aim of enhancing the anti-tumor immune response beyond anti-PD1/PD-L1 drugs [115].

The TNBC microenvironment is characterized by the pro-inflammatory cytokines and chemokines and alternative immunotherapy strategy based on the silencing of cytokine receptors represents an intriguing field of research [116]. In this context, the deregulation of Multiple Copies in T-cell Malignancy 1/miR-34a/interleukin-6/interleukin-6 Receptor (MCT-1/miR-34a/IL-6/IL-6R) signaling axis, involved in epithelial-mesenchymal transition, M2-like macrophages polarization and TNBC cell invasiveness, has been identified as a potential therapeutic target. Pre-clinical trials showed interesting results by the use of MCT-1 inhibitor in combination with anti-IL-6/IL-6R monoclonal antibody, tocilizumab, or with short/small hairpin RNA, thus allowing further investigations [117].

Moreover, a synergistic effect by the addition of targeted therapies (PI3K, MEK, VEGF, EGFR and PARPis), that are able to reinforce immune microenvironment, to ICI is expected and currently under evaluation in ongoing clinical trials. Preliminary positive data derived also from trials exploring novel immune system modulators in TNBCs as well as adaptive cell therapy, chimeric antigen receptor (CAR) T cell therapy and vaccines [118].”

We hope that we met your requirements fully.

Best regards,

Dr. Anna Diana

(On behalf of all authors)

Reviewer 2 Report

This review is well summarized and well written. I think this review is going to help readers to understand the current landscape of TNBC treatment, focusing on the early-stage cancers. 

There are a few areas /items we can implement to improve this manuscript.

1) The classification of TNBC: molecular subtypes and gene signature is a well-summarized section. however, as an expert review, I strongly encourage the authors to suggest a summary recommendation - how do authors envision this discrepancy to be resolved? what could be the implementation of the biomarker to test this subtype in the clinic? What would be the recommendation of day-to-day practice in today's clinic?

2) This statement is somewhat controversial. This is because the standard care is not currently designed/changed based on the TIL count. I advise revising this statement until we have new and accumulated data regarding this.

In conclusion, TILs count is considered a prognostic biomarker in TNBC (level of evidence I) and it should be routinely assessed in clinical practice, following TILs Working Group 195 recommendation

3) If the word count allows, I also recommend the authors to add the section to describe the clinical trials to utilize/target molecular profiling in the neoadjuvant treatment- there are several that we plan different combination therapy using targeted therapeutics, based on personalized molecular profiles. 

4) Given this review aims for the 'early TNBC', I believe it is only fair to include 5-FU based therapy to be included after the platinum - either neoadjuvant or adjuvant. 

Please address above, and the manuscript will become more comprehensive and informative. 

Author Response

Dear Reviewer, we would like to thank you for careful and thorough reading of our manuscript titled “Early Triple Negative Breast Cancer: conventional treatment and emerging therapeutic landscapes”. We are sure that your suggestions will make more informative this review article.

Response to Reviewer 2:

This review is well summarized and well written. I think this review is going to help readers to understand the current landscape of TNBC treatment, focusing on the early-stage cancers. There are a few areas /items we can implement to improve this manuscript.

Point 1. The classification of TNBC: molecular subtypes and gene signature is a well-summarized section. however, as an expert review, I strongly encourage the authors to suggest a summary recommendation - how do authors envision this discrepancy to be resolved? what could be the implementation of the biomarker to test this subtype in the clinic? What would be the recommendation of day-to-day practice in today's clinic?

Response 1: We entered some suggestions on how to overcome the discrepancy between the different molecular classifications, with reference to therapeutic implications that might arise. Furthermore, we added some recommendations in day-to-day practice to select personalized treatment for TNBC patients.

Page 5, lines 95-111. “Despite efforts in classifying TNBC, the discordance in the number of molecular subtypes and methods used as well as intratumor genomic heterogeneity represented a barrier in developing targeted therapies. However, by reviewing all genomic studies, emerged that TNBC consists of four main subgroups: BL, Mesenchymal, LAR and immune-enriched. This simplified molecular portrait of TNBC provided the basis for a new subtype-driven therapeutic approach. For example, in BL cancers, targeting DNA damage response pathways by using platinum salts and PARP inhibitors (PARPis) could be an effective therapeutic strategy while the Mesenchymal subtype would seem to be more sensitive to PI3K, Vascular Endothelial Growth Factor 2 (VEGFR2), Fibroblast Growth Factor Receptor (FGFR) or mammalian target of rapamycin (mTOR) inhibitors [10]. Several phase I-II trials have suggested the activity of anti-androgen agents in LAR subgroup, even if results from confirmatory phase III studies are needed to validate the efficacy of this strategy [11]. Furthermore, ongoing trials are investigating the usefulness of immunotherapy for treatment of patients affected by immune-enriched TNBC.

In conclusions, the implementation of Next generation sequencing and other genomic assays into everyday practice is expected soon to deliver personalized treatment for TNBC, but, until then, patients’ enrolment in clinical trials that stratify according to the presence of a specific gene signature should be strongly encouraged”.

Point 2. This statement is somewhat controversial. This is because the standard care is not currently designed/changed based on the TIL count. I advise revising this statement until we have new and accumulated data regarding this. In conclusion, TILs count is considered a prognostic biomarker in TNBC (level of evidence I) and it should be routinely assessed in clinical practice, following TILs Working Group 195 recommendation

Response 2:    I agree. Although TILs count represents a validated prognostic marker, it cannot be use to guide therapeutic choices. Therefore, we rephrased as follow.

Page 7, lines 217-219. “In conclusion, TILs count is considered a well-established prognostic biomarker in TNBC (level of evidence I) and its clinical routine use is recommended by the 2019 Saint Gallen Consensus. However, to date, TILs scoring should not be used to guide neo-(adjuvant) treatment choices [44]”.

Point 3. If the word count allows, I also recommend the authors to add the section to describe the clinical trials to utilize/target molecular profiling in the neoadjuvant treatment- there are several that we plan different combination therapy using targeted therapeutics, based on personalized molecular profiles. 

Response 3: Unfortunately, words count doesn’t allow to add a dedicated section (we have already exceeded those allowed by over 1000 words). However, we added the following sentences addressing your point. In particular, we pointed out the intriguing rationale of combined therapy by using targeted agents and immunotherapy. Furthermore, we reported the presence of ongoing neoadjuvant trials that stratify patients on the basis of specific gene signature in order to facilitate the development of biomarker-driven therapies.

Page 17-18, lines 593-602. “Moreover, a synergistic effect by the addition of targeted therapies (PI3K, MEK, VEGF, EGFR and PARPis), that are able to reinforce immune microenvironment, to ICI is expected and currently under evaluation in ongoing clinical trials. Preliminary positive data derived also from trials exploring novel immune system modulators in TNBCs as well as adaptive cell therapy, chimeric antigen receptor (CAR) T cell therapy and vaccines [118].

Finally, in the transcriptomic era, the identification of multiple potentially actionable targets led to categorize TNBC into smaller and smaller subgroups. In this scenario, several biomarker-driven neo-adjuvant trials are investigating the efficacy of targeted therapies stratifying patients before randomization accordingly to distinct gene signature, with the aim to select specific population to include in confirmatory phase III studies”.

Point 4.  Given this review aims for the 'early TNBC', I believe it is only fair to include 5-FU based therapy to be included after the platinum - either neoadjuvant or adjuvant. 

Response 4: We inserted the results of a recent meta-analysis that explored the efficacy of capecitabine as additional treatment or given instead of another therapy in a large cohort of BC patient. Furthermore, we discussed on the lack of evidence about the addition of adjuvant capecitabine after neo-adjuvant platinum and informed about an ongoing study that is comparing capecitabine with platinum agents in adjuvant setting. As follows:

Page 9, lines 320-329. “Finally, a recent meta-analysis of 12 prospective randomized trials including more than 15,000 BC patients reported that only TNBC subset derived significant survival benefit from adjuvant capecitabine, particularly when it was given in addition to systemic treatment rather than instead of another therapy [68].

However, the role of adjuvant capecitabine in TNBC patients who received platinum-based chemotherapy and/or ICI, as part of neoadjuvant strategy, remains to be investigated. Moreover, the results of ongoing phase III trial comparing capecitabine with platinum salts in TNBC with residual disease after NACT are awaited (NCT02445391).

In clinical practice, the addition of adjuvant capecitabine should be considered in TNBC patients, especially in those with residual disease after standard NACT”.

We hope that we met your requirements fully.

Best regards,

Dr. Anna Diana

(On behalf of all authors)

Reviewer 3 Report

A comprehensive review.

 Two points require accuracy:

1) There are still  conflicting results regarding the predictive role of HRD.

For example, Tutt's et al , (Nature Medicine, 2018) reported a negative study

2) The quotation (No. 50) that 40 % of the patients will  relapse is referring to patients who had received neoaduvant treatment and not to all patients.

Author Response

Dear Reviewer, we would like to thank you for careful and thorough reading of our manuscript titled “Early Triple Negative Breast Cancer: conventional treatment and emerging therapeutic landscapes”. 

Response to Reviewer 3:

A comprehensive review. Two points require accuracy:

Point 1. There are still conflicting results regarding the predictive role of HRD. For example, Tutt's et al, (Nature Medicine, 2018) reported a negative study

Response 1: We added the negative results reported in metastatic setting and rephrased the period as follow.

Page 6, lines 159-163. “All these studies proved a strong correlation between HRD status and responsiveness to platinum-based NACT among TNBC patients. Noteworthy, in contrast with these data, platinum demonstrated no additional benefit compared to docetaxel in BRCAness metastatic TNBC, suggesting that its activity is not epigenetically driven by HRD status [20]”.

Point 2. The quotation (No. 50) that 40 % of the patients will relapse is referring to patients who had received neoadjuvant treatment and not to all patients.

Response 2: Yes. The percentage of 40% has been removed because we are referring to all TNBC population.

Page 8, lines 280-282. “The therapeutic decision-making process represents a crucial point for early TNBC due to its higher propensity to relapse rapidly, generally within 2-3 years from diagnosis, with limited treatment options available in the metastatic setting beyond conventional cytotoxic drugs”.

We hope that we met your requirements fully.

Best regards,

Dr. Anna Diana

(On behalf of all authors)